# Feed Clusters According to In Situ and In Vitro Ruminal Crude Protein Degradation

**DOI:** 10.3390/ani13020224

**Published:** 2023-01-07

**Authors:** Paul Okon, Martin Bachmann, Monika Wensch-Dorendorf, Natascha Titze, Markus Rodehutscord, Christiane Rupp, Andreas Susenbeth, Jörg Michael Greef, Annette Zeyner

**Affiliations:** 1Institute of Agricultural and Nutritional Sciences, Martin Luther University Halle-Wittenberg, 06120 Halle (Saale), Germany; 2Institute of Animal Science, University of Hohenheim, 70599 Stuttgart, Germany; 3Institute of Animal Nutrition and Physiology, Kiel University, 24118 Kiel, Germany; 4Julius Kühn Institute, Federal Research Centre for Cultivated Plants, Institute for Crop and Soil Science, 38116 Braunschweig, Germany

**Keywords:** concentrates, grass silages, ruminal crude protein degradation, in situ, in vitro, *Streptomyces griseus* protease

## Abstract

**Simple Summary:**

The objective of the present study was to assess the suitability of an enzymatic laboratory method to estimate ruminal protein degradation. In situ data were used as reference. Appropriate in vitro methods are important to overcome methodological and ethical shortcomings, associated with the use of experimental animals. A cluster analysis was performed on the basis of differences between in vitro and in situ protein degradation. Among the 40 feedstuffs we tested, this difference was lowest in legume grains and highest in cereal by-products and barley. The feedstuffs clustered unspecific, not relatable to nutrient composition, origin or treatment. However, it was often obvious that additional carbohydrate-degrading enzymes should be used to assist the laboratory method, based solely on protease, to make it more conform with the in situ reference data.

**Abstract:**

Effective degradation (ED) of crude protein (CP) was estimated in vitro at 0.02, 0.05 and 0.08 h^−1^ assumed ruminal passage rates for a total of 40 feedstuffs, for which in situ ED was available and used as reference degradation values. For this, the *Streptomyces griseus* protease test was used. The differences between in vitro CP degradation and the in situ CP degradation values were lowest in legume grains and highest in cereal by-products and barley. The differences between in situ and in vitro ED were expressed using a degradation quotient (degQ), where degQ = (ED_in vitro_ − ED_in situ_)/ED_in situ_. Among the tested feedstuffs, eight specific clusters were identified according to degQ for the assumed passage rates. The feedstuffs clustered in an unspecific way, i.e., feedstuffs of different nutrient composition, origin or treatment did not necessarily group together. Formaldehyde–treated rapeseed meal, soybean meal, wheat, a treated lupin, sunflower meal and barley could not be assigned to any of the clusters. Groupwise degradation (range of degQ for assumed passage rates are given in brackets) was detected in grass silages (−0.17, −0.11), cereal by-products together with sugar beet pulp (−0.47, −0.35) and partly in legume grains (−0.14, 0.14). The clustering probably based on different specific nutrient composition and matrix effects that influence the solubility of feed protein and limit the performance of the protease. The matrix can be affected by treatment (chemically, thermally or mechanically), changing the chemical and physical structure of the protein within the plant. The *S. griseus* protease test had reliable sensitivity to reflect differences between native feedstuffs and treatments (thermally or chemically) that were found in situ. The in situ results, however, are mostly underestimated. The clustering results do not allow a clear conclusion on the groupwise or feed-specific use of carbohydrate-degrading enzymes as pre- or co-inoculants as part of the *S. griseus* protease test and need to be tested for its potential to make this test more conform with in situ data.

## 1. Introduction

The estimation of ruminal crude protein (CP) degradation is an essential part of feed protein evaluation, with feed protein being a limited and expensive nutrient source. Therefore, sufficient quantification of ruminal CP degradation is of high interest in order to assess nitrogen utilization efficiency of ruminant livestock to meet precisely the animal’s requirement. The degradation of CP which is measured in vivo has been considered as reference. However, this method is laborious, time-consuming and associated with errors due to variation among individual animals and use of markers [1,2,3]. Therefore, in situ determination of ruminal CP degradation is widely used as reference [4]. The use of in situ data is often critically discussed, especially in terms of repeatability of the results. There is, however, a reliable methodological protocol existing that ensures repeatability [5]. This protocol recommends, for example, the correction of in situ degradation data for the microbial nitrogen that is synthesized during the incubation of feeds in the rumen [6]. For future applications, it seems worthwhile to search for methods that do not rely on cannulated animals and are potentially useful for routine analysis [7]. A purely enzymatic in vitro method is the *Streptomyces griseus* protease test, which was developed by Krishnamoorthy et al. [8]. Protease from the bacterial species *S. griseus* has a broad activity spectrum and may hydrolyze proteins (i.e., oligopeptides) up to 90% [9,10]. Its high reactivity comes from endo- and exopeptidases, especially metalloendopeptidase activity [10,11]. Licitra et al. [10] indicated the ratio of protease to true protein (TP) concentration has influence on CP degradation and standardized the method to that effect. Moreover, incubation times were referenced to type of feed and feed characteristics [10,12]. Several studies have shown close agreement between CP degradation estimated in situ or using the *S. griseus* protease test both in concentrates and roughages [7,13,14,15,16]. Feed-specific degradation kinetics and effective degradation of protein (ED) at different passage rates have been barely described. A large part of feed protein is associated to carbohydrates, i.e., starch and fiber, as a kind of matrix that influences the degradation capability of proteases. Such matrix effects could be responsible for the inability of protease to degrade the entire feed protein [17,18,19].

We have evidence from a pilot study that the reliability of the *S. griseus* protease test estimating ED by the measure of in situ ED strongly depends on the incubated feedstuff and clusters may be defined that rely on feed or treatment characteristics [20]. From this, we conclude that the efficiency of the *S. griseus* protease may be influenced by matrix effects, which lead to protein degradation of feedstuffs clusters according to similar nutrient characteristics.

The objective of the present study was to assess the suitability of the *S. griseus* protease test for estimating ED of CP from 40 feedstuffs using the in situ test as a reference method.

Our hypothesis was that specific characteristics (e.g., nutrient content, treatment) of individual feedstuffs or groups of feedstuffs lead to a differentiation with regard to the susceptibility of the feed protein to protease, and thus, to specific clustering.

## 2. Materials and Methods

### 2.1. Feedstuffs and Treatments

A set of 40 different feedstuffs for which in situ CP degradation data were available has been used for the in vitro investigations to obtain a wide range of different feedstuffs. This set contained soybeans, soybean meal (SBM), sunflower meal (SFM), barley and wheat grains, wheat bran, corn gluten feed (CGF), sugar beet pulp (SBP) and dried distillers’ grains with solubles (DDGS). In addition, some feedstuffs were subjected to treatment (Table 1). Lupin grains of cultivars *Boregine* and *Boruta* were tested, both native and treated. Additionally, six differently treated Rapeseed meals (RSM) were investigated indicated by letters a to d (Table 1). Two RSM, RSMc and RSMd, were tested native and treated as described in Table 1. With exception of the over-toasted RSM (RSMb), all further treated RSM (RSMa, RSMc and RSMd) were provided by industry, and specific treatment information was not available.

Three different cultivars of peas were investigated: *Hardy*, *Astronaute* and *Navarro*. As described by Rupp et al. [22], perennial ryegrass (*Lolium perenne*) was cut in 2017, wilted at 40 °C under temperature control and chopped to 20–30 mm particle size. A total of 16 grass silages were made from this material (90 d at 25 °C in glass jars), for which eight wilting stages were produced (I: 170 g dry matter (DM), II: 310 g DM, III: 390 g DM, IV: 420 g DM, V: 470 g DM, VI: 530 g DM, VII: 580 g DM and VIII: 600 g DM/kg) and ensiled with or without adding a mixture of homo- and heterofermentative lactic acid bacteria. Information on the ensiling process and silage quality parameters is given by Rupp et al. [22]. Nutrient concentrations of all feedstuffs are summarized in Table 2.

### 2.2. Origin of In Situ Data

Animal experiments were not part of this study because all in situ data originated from preliminary studies conducted under approval no. V319/14 TE. Ruminal CP degradation of concentrates was determined at the Institute of Animal Science, University of Hohenheim using a standardized assay [22,25,26,27,28]. In brief, feedstuffs were incubated in the rumen of rumen-fistulated Jersey cows and three cows were used for each feedstuff. Incubations were made in polyester bags (Ankom Technology, Macedon, New York, USA) with a pore size of 50 µm (30 µm for RSM) and internal dimensions of 5 × 10 cm, 10 × 20 cm and 11 × 22 cm for a time period of up to 72 h and in case of SFM and all pea cultivars for a time period of up to 48 h. A minimum of three bags was used of each point in time and cow and contents after incubation were pooled for chemical analysis. The in situ degradation data of RSMa, native and treated RSMc and RSMd were not published yet. In situ protein degradation was expressed as a percentage of CP at each specific incubation time.

### 2.3. In Vitro Incubations

The *S. griseus* protease test was performed according to Licitra et al. [10]. The feedstuffs were ground through 1 mm sieve size using a standard laboratory sample mill. Briefly, 0.5 g were weighed in 50 mL centrifuge tubes, filled with 40 mL of borate-phosphate buffer (12.20 g NaH_2_PO_4_ × H_2_O + 8.91 g Na_2_B_4_O_7_ × 10 H_2_O/L with pH 6.7–6.8) and placed into a drying oven for 1 h at 39 °C as pre-incubation. The protease solution contained 0.58 U of nonspecific type XIV *S. griseus* protease (Sigma-Aldrich Chemie GmbH, Munich, Germany) per mL and was added after 1 h pre-incubation at a ratio of 24 U/g TP. The concentration of TP was calculated according to the Cornell Net Carbohydrate and Protein System (CNCPS) as CP minus non-protein nitrogen (fraction A) [23]. Samples of incubation time 0 h were taken immediately after pre-incubation without addition of protease solution. Subsequently, the feedstuffs were incubated in duplicate for 2, 4, 6, 8, 16 and 24 h, respectively. Afterwards, sample tubes were filtered through Whatman #41 filter circles and rinsed out with 150 mL distilled water each. The filters were air-dried overnight, and nitrogen was analyzed in the residues and blank filters using a FOSS Kjeltec^TM^ 8400 unit (Foss GmbH, Hamburg, Germany). This procedure was repeated a minimum of four and a maximum of six times to obtain at least four replicates for each feedstuff. Concentrations of rumen undegraded protein (RUP) were determined according to Bachmann et al. [29] as follows (considering a sample weight of 0.5 g):RUP (g/kg DM) = ((N_residue_ × 6.25 × 10)/(0.5 × DM_feed_)) × 10,
where N_resdiue_ is the nitrogen measured in filter residues (mg) corrected by blank filters and DM_feed_ is the DM content of feedstuff (%). Degraded protein (% of CP) was considered the reciprocal of RUP at each specific incubation time.

### 2.4. Effective Protein Degradation

The following calculations were made using SAS 9.4 (SAS Institute Inc., Cary, NC, USA). In a first step, the in situ CP degradation data of the tested substrates were reanalyzed by fitting CP degradation (as % of CP) measured after 0, 2, 4, 6, 8, 16, 24, 48 and, if applicable, for 72 h of incubation to the exponential equation provided by McDonald [30] and Steingass and Südekum [31] using the MODEL procedure of SAS 9.4. To describe CP degradation, the washout protein *a*, which instantly disappears at time *t* = 0, *b*, which is the protein potentially degradable in the rumen, and *c*, which is the degradation rate of fraction *b*, were estimated. The possible appearance of a discrete lag phase *L*, at which no ruminal degradation occurs, was considered using a broken-line approach. As long as *t* ≥ *L*, CP degradation was fitted to the regression function, whereas if *t* < *L*, CP degradation was considered to be equal to *a*. The estimates of the lag phase were set to be greater or equal to zero; *a* + *b* was restricted to be lower or equal 100%. Note that the slightly different methodological approach led to ED estimates which were somewhat different from those previously published [22,25,26,27,28] using partly the same feedstuffs as in the current study. In a second step, the in situ CP degradation data were corrected for the amount of microbial nitrogen present in the feed residues at each specific incubation time using the equations of Parand and Spek [6] as recommended by GfE [5]. For this, feedstuffs were grouped by roughages, concentrates and low protein concentrates (CP < 300 g/kg DM) and the specific equations provided by Parand and Spek [6] were applied. The potential contamination with microbial nitrogen over time is summarized in Appendix A. In a third step, in situ CP degradation was estimated once with 72 h or 48 h maximal incubation time and once with maximal incubation time reduced to 24 h in accordance with maximal in vitro incubation time, and all estimations were repeated both without and with correction upon microbial nitrogen. The in situ dataset comprised three replicates per feed sample (i.e., three animals). The in vitro CP degradation was analyzed analogously with a maximal incubation time of 24 h. Within the in vitro dataset, outliers were identified using boxplots and eliminated. Outliers were defined as observations more far than three times of interquartile range. The in vitro dataset comprised six replicates (i.e., six runs) and a minimum of three replicates after elimination of outliers. Effective CP degradation, either in situ or in vitro, was calculated on the basis of the estimated parameters *a*, *b*, *c*, and *L* as described by Wulf and Südekum [32] for assumed ruminal passage rates of 0.02 (ED_2_), 0.05 (ED_5_) and 0.08 h^−1^ (ED_8_). The whole investigation was conducted under the assumption that the degradation data obtained by the in situ method is the reference to which in vitro ED was compared. The SAS script used for all calculations can be obtained from the authors on request.

### 2.5. Chemical Analyses

Concentrations of DM, crude nutrients and detergent fiber fractions were analyzed according to the Association of German Agricultural Analytic and Research Institutes [33] using methods no. 3.1 (DM), 4.1.1 (CP), 5.1.1 B (AEE), 6.5.1 (aNDFom), 6.5.2 (ADFom) and 8.1 (CA). Neutral detergent fiber was determined after amylase treatment. Neutral detergent fiber and acid detergent fiber were expressed exclusive of residual ash. Starch was determined using the amyloglucosidase method (VDLUFA, 2012; method no. 7.2.5) similar to Grubješić et al. [25] and enzymatically according to Seifried et al. [34]. The CNCPS protein fraction A (non-protein nitrogen) was determined according to Licitra et al. [23] and in grass silages according to Higgs et al. [24]. All measurements of nitrogen were performed using the Kjeldahl method.

### 2.6. Statistical Analysis

Statistical analysis was performed using SAS 9.4. The effects of a correction of in situ CP degradation for percentages of microbial nitrogen contained in the feed residues and a reduction of maximal incubation time (*t* = 72 h/48 h or *t* = 24 h) on ED_2_, ED_5_ and ED_8_ were tested using the NPAR1WAY procedure and Kruskal–Wallis test. Differences between in situ and in vitro estimates of ED_2_, ED_5_ or ED_8_ for both 24 and 72 h maximal incubation times (in situ) were tested using pooled *t*-test or the Satterthwaite approximation of the *t*-test if applicable according to folded *F*-test. In cases where Gaussian distribution of the studentized residuals was not given, we used the Wilcoxon rank sum test with the NPAR1WAY procedure. Differences between in situ and in vitro estimates of ED_2_, ED_5_ or ED_8_ in native or treated feed samples were tested by pooled *t*-test. Homogeneity of sample variances and Gaussian distribution of the studentized residuals were confirmed. For all tests, statistical significance was given with *p* < 0.05. To compare effective CP degradation between the two different methodologies (in situ vs. in vitro), differences between in situ and in vitro ED_2_, ED_5_ and ED_8_ were additionally expressed as a degradation quotient (degQ). The degQ was calculated as follows:degQ = (ED_in vitro_ − ED_in situ_)/ED_in situ_.

Clustering was examined by single linkage method separately for every feedstuff and for assumed ruminal passage rates of 0.02 h^−1^, 0.05 h^−1^ and 0.08 h^−1^ including degQ or including only the concentrations of crude nutrients, detergent fibers and starch. Missing data of starch concentration in feedstuffs were provided by DLG [35]. Grass silages, SBP, native and treated RSMc and RSMd did not contain starch and soybeans contained 58 g starch/kg DM. A dendrogram was created showing the clusters.

## 3. Results

The estimated amount of microbial nitrogen present in feed residues is listed in Appendix A. Microbial nitrogen was highest in grass silages (55.8–58.5% of total nitrogen) and lowest in the native lupin *Boregine* (5.1% of total nitrogen) after 72 h incubation time. The effects of a correction for microbial nitrogen contamination and a reduction of maximal incubation time (72 h or 48 h to 24 h) on in situ ED_2_, ED_5_ and ED_8_ are shown in Appendix A. Irrespective of the incubation time (72 h or 48 h or 24 h), correction for microbial nitrogen elevated in situ ED which reached significance in wheat grain with up to 2% points, in RSMa (expander-treated) and in RSMb (over-toasted) with up to 5% points and in all grass silages with up to 6% points (*p* < 0.05). The reduction of incubation time to maximal 24 h had mostly no or merely a small effect on in situ ED; only in RSMd (formaldehyde-treated and nitrogen-uncorrected), ED_2_ was reduced by a maximal of 11% points. However, reduction of incubation time from 72 h or 48 h to 24 h affected ED of protein and comparison to in vitro ED (Appendix A) far less than the correction for microbial nitrogen contamination (Appendix A). On that basis, in situ CP degradation over maximal 72 h corrected for microbial nitrogen contaminations was used as reference for comparison of in vitro results reported in the following (Table 3).

Reliable estimation of ED in vitro was not possible in case of faba beans and corn due to implausible estimates of CP degradation parameters (Appendix A). Effective CP degradation was mainly underestimated using the *S. griseus* protease test (by maximal 48% points; *p* < 0.05; Table 3), which is shown by negative quotients (Table 4). Only in the treated lupins *Boregine* and *Boruta*, the pea *Navarro* and the SFM at a passage rate of 0.08 h^−1^ and in the native lupins *Boregine* and *Boruta* and SBM at 0.05 and 0.08 h^−1^ passage rates, ED was higher in vitro than in situ (Table 3). Regardless of ruminal passage rate, the largest differences between in situ and in vitro CP degradation were found in barley grains and industrial by-products (DDGS, CGF, wheat bran and SBP). In these feedstuffs, *a*, *b* or *c* were underestimated up to 53% points by the in vitro method (*p* < 0.05). Significant differences between in situ and in vitro estimates were also found in oilseeds following fat extraction and other processing processes (*p* < 0.05). In legume grains (with exception of faba beans), in situ and in vitro estimates agreed well, although some differences were significant (*p* < 0.05).

The calculated degQ for the assumed passage rates showed that lupins and pea grains had nearly degQ of 0, whereas by-products and barley grain had the lowest degQ of less than −0.50 (Table 4).

The cluster analysis including crude nutrient, detergent fiber and starch concentrations of the feedstuffs clearly showed eight clusters (cluster 1: native and treated variants of RSMc and RSMd; cluster 2: treated variants of lupins *Boregine* and *Boruta*; cluster 3: over-toasted RSM (RSMb) and SFM; cluster 4: expander-treated RSM (RSMa) and DDGS; cluster 5: all grass silages; cluster 6: wheat bran and CGF; cluster 7: pea varieties *Astronaute* and *Hardy*; cluster 8: wheat and barley). Soybeans, SBM, native variants of lupins *Boregine* and *Boruta*, SBP and the pea *Navarro* were arranged outside of any cluster (Appendix A).

Separate inclusion of degQ at 0.02 h^−1^, 0.05 h^−1^, 0.08 h^−1^ and all degQ together revealed that 37 clusters appeared (Figure 1, Appendix A). The grass silages, native variants of RSMc and RSMd, SBP, CGF, wheat bran and DDGS clustered together irrespective of passage rate. The lupins were combined with the peas in varying cluster combinations. The other RSM, SFM and wheat clustered diffusely in varying combinations with other feedstuffs. Some feedstuffs, however, were not attributed to any cluster: for degQ at 0.02 h^−1^, formaldehyde-treated RSM (RSMd), SFM and barley; for degQ at 0.05 h^−1^, SBM and barley, for degQ at 0.08 h^−1^, SBM, over-toasted RSM (RSMb), wheat bran, the native lupin *Boregine* and barley; and for all degQ together, formaldehyde-treated RSM (RSMd), SBM, over-toasted RSM (RSMb), treated lupin *Boruta* and barley (Figure 1 and Figure 2).

In Figure 3, all feedstuffs were arranged in the same order as in Figure 2. A total of eight clusters were identified and were delimited by dashed lines (cluster 1: soybeans and expander treated RSM (RSMc); cluster 2: native variants of RSMc and RSMd; cluster 3: all grass silages and expander-treated RSM (RSMa); cluster 4: pea varieties *Navarro* and *Astronaute*; cluster 5: pea variety *Hardy* and treated variant of lupin *Boregine*; cluster 6: native variants of lupin *Boruta* and *Boregine*; cluster 7: wheat bran and DDGS; cluster 8: SBP and CGF) (Figure 2, Appendix A).

Within the in situ dataset, significant differences in ED were found between native and treated feedstuffs in most of the comparisons (Appendix A). These differences between native and treated feedstuffs were likewise obtained using the *S. griseus* protease test (Figure 4).

## 4. Discussion

Protease from *S. griseus* has widely been used for estimation of ruminal CP degradation [7,13,16,36]. As Edmunds et al. [7] described, comparison among studies is difficult, because either enzyme concentration or incubation conditions differ. The standardized protocol of Licitra et al. [10] is, therefore, a basis on which the *S. griseus* protease test can be performed under defined conditions. In accordance with Cone et al. [36] and Cone et al. [37], in this study, we measured degradation of CP at 0, 2, 4, 6, 8, 16 and 24 h, which allowed displaying specific degradation kinetics and to compare them with in situ results. Although degradation values from in vivo studies are considered to be the best possible reference, they are almost not available. The results of the *S. griseus* protease test were, therefore, compared to in situ degradation values as the best available reference. It should be noted that the in situ method is associated with relevant uncertainties (i.e., microbial attachment and particle losses), which is why the results are subjected to variability and bias [38]. These limitations highlight the potential of in vitro methods in terms of standardizable, reproducible and inexpensive methods for estimating ruminal protein degradation.

As a first step, we examined the impact of nitrogen from increasing adherence of microbial biomass to the feed residues during in situ incubation [6,39], and secondly of the reduction of the incubation time on in situ predictions of ED_2_, ED_5_ and ED_8_. Microbial nitrogen adhering to in situ feed residues was estimated according to Parand and Spek [6]. As shown previously [6,39], microbial nitrogen contamination of feed residues during ruminal in situ incubation is substantially lower in concentrates than in roughages. We calculated a maximal contamination with microbial nitrogen of 58% of total nitrogen in feed residues after 72 h of ruminal in situ incubation with lower values for concentrates (5–45% of total nitrogen) and higher values for grass silages (55%–58% of total nitrogen) as the only forage source in the current study. The correction for microbial nitrogen contamination elevated the estimated ED_2_, ED_5_ and ED_8_ in all tested feedstuffs. The correction for microbial nitrogen resulted in greater differences between in situ and in vitro estimated ED especially in grass silages and SBP and is, therefore, deemed to be relevant in at least some of the concentrates, and especially in roughages.

For routine applications, a simple and timely affordable in vitro test for the determination of CP degradation of feeds is required. Using *S. griseus* protease, incubation times of maximal 30 h are thought to be reliable for concentrates, by-products from food processing and forages [10]. Although in our re-calculation of in situ data, the reduction of incubation time in situ from 72 h or 48 h to 24 h had just a small effect on ED_2_, ED_5_ and ED_8_ (Appendix A), we followed the assumption of Steingass and Südekum [31] and recommendations by GfE [5] that the incubation time of at least 48 h is necessary for the reliable estimation of ED.

The rumen is colonized by a diverse commensal microbiota consisting of bacteria, protozoa, and anaerobic fungi [40]. Among bacteria, the most intensively studied group of rumen microbes, *Prevotella* was the most dominant genus found in ruminal fluid [41,42]. They are closely associated with protein and carbohydrate degradation [42,43] and may also act cellulolytic or synergistically with other cellulolytic microorganisms [42,44]. Rate and extent of CP degradation largely depend on the proteolytic activity of ruminal microflora and feed protein composition [45], but also amylolytic and cellulolytic activities of the microbes support ruminal degradation of proteins [17,18,45,46]. This might cause the gap between in situ estimates and those obtained using *S. griseus* protease as a sole agent.

Degradation of feed protein is mainly influenced by its solubility. In grass silages and legume grains, soluble protein was highest, whereas it was lowest in RSM and soybeans (Table 2). The proportion of washout protein (*a*) (in situ) or the protein soluble in borate-phosphate buffer (in vitro) plays an essential role especially in legume grains (with exception of faba beans). Hedqvist and Uden [47] determined the highest proportion of buffer soluble nitrogen (B1) in pea grains, lupin grains and grass silages. The protein solubility is influenced by the native protein composition, i.e., the distribution of prolamins, glutelins, albumins and globulins [48,49]. Most important, however, is the localization of proteins. The plant protein is structurally enclosed in the matrix composed of cellulose, hemicellulose and pectin or associated to starch and have to be dissolved prior to efficient CP degradation [45]. The cereal by-products (wheat bran, CGF and DDGS) are enriched by plant cell wall constituents (aleurone and pericarp). These parts of the grain comprise cell wall associated proteins [50,51]. More than 50% of total protein of wheat bran, CGF and DDGS is associated to fiber and is, therefore, less accessible to proteases [50,52]. These feedstuffs and SBP had the lowest degQ compared to the oilseed by-products (RSM, SFM and SBM). Pedersen et al. [19] found protein solubilization in DDGS to be increased by up to 31% following the addition of xylanase. The combination of xylanases and protease had the greatest potential to degrade non-starch polysaccharides, such as arabinoxylan, and release nutrients from DDGS [19]. The fiber-protein matrix can be influenced by heating and chemical treatment during food/feed processing. The large number of treatment options, i.e., the combination of time, temperature, use of water and reducing substances, results in a wide range of differently processed feedstuffs [53,54]. As a result of processing, the protein as a component of the matrix is structurally and chemically altered [54,55,56,57]. During heating and chemical treatment, the resistance to proteolysis might increase by denaturation of protein and/or formation of Maillard reaction products [54,58,59,60,61]. Effective CP degradation of by-products was considerably underestimated in vitro compared to the in situ results. The reduced in vitro ED can be attributed to separation processing, which leads to products enriched in cell wall-associated proteins (wheat bran, DDGS and CGF) or to physical or chemical post production treatments, as in DDGS, RSM, SFM, SBM and SBP.

Proteins localized in grains (cereal grains and legume grains) are associated to starch and this may influence protein solubility by interactions between protein and starch. These proteins surround or encapsulate starch granules and are a physical barrier to starch digestion [62]. In corn, in situ starch degradation was negatively correlated with the CP concentration of the grain [34]. In the endosperm, embedding of starch granules occurs in highly variable spherical structures which depend on source, genotype and environmental conditions. This affects the vulnerability to enzymes [63,64,65]. Assoumani et al. [18] reported increased CP degradation through *S. griseus* protease with the addition of amylase and ß-glucanase in feeds with more than 23% starch on DM basis (corn, wheat and barley). They attributed the smaller differences between in situ and in vitro CP degradation to improved accessibility or vulnerability of protease to the protein matrix. Literature data revealed that toasting may decrease degradation of protein in lupins [26,58,59]. Bachmann et al. [66] showed, on the basis of scanning electron micrographs, that in pea grains, heat treatment led to an alteration of the protein matrix. Then, heat treatment limits proteolysis of the matrix surrounding starch granules. This is probably an effect of Maillard reactions or the inactivation of trypsin inhibitor activity [67,68].

Full fat soybeans, as another example, are characterized by large differences between in situ and in vitro ED estimates as well. Guillamón et al. [69] reported that soybeans contain high levels of trypsin inhibitors, which can inhibit protease activity. Another reason could be the high concentration of AEE influencing the activity of protease.

In forages, it was found that the *S. griseus* protease test had good accuracy with estimating ruminal CP degradation probably due to a high protein solubility [7,15,37]. In grass silages, specifically, proteolysis during ensiling increased the proportion of non-protein nitrogen, whereby protein is released from the fiber matrix [7]. Despite their high solubility, the grass silages in the present study had markedly lower in vitro ED compared to in situ ED. Abdelgadir et al. [17] used fibrolytic enzymes prior to the incubation with protease, which reduced differences between in situ and in vitro degradation of CP from alfalfa and meadow hay. The grass silages mostly had a degQ between −0.1 and −0.2, which was very uniform both among variants and among ED_2_, ED_5_ and ED_8_. Prospectively, this makes mathematical correction of in vitro estimates possible and does not necessarily require modification of the test.

Our hypothesis was that specific characteristics (i.e., nutrient content or treatment) of individual feedstuffs or groups of feedstuffs lead to feed clusters with regard to the susceptibility of the CP to protease. On the basis of the selected feeds, clusters could be identified with regards to the degQ at 0.02, 0.05 and 0.08 h^−1^ assumed ruminal passage rates. The inclusion of all degQ resulted in eight clusters (Figure 3). Other feedstuffs could not be assigned to any cluster. Some clusters were characterized by similar nutrient compositions of included feedstuffs (clusters 2, 4 and 6), others by partial (cluster 1) or very clear differences (clusters 3, 5, 7 and 8). Such differences can be attributed to treatment effects on feed protein. The clustering of degQ clearly shows diffuse assignment of untreated with treated feedstuffs in common clusters (clusters 1, 3 and 5) or individually outside of any cluster (SBM, SFM and treated lupin *Boruta*) (Figure 3). Especially, the aggressive treatments of RSM could contribute to separate arrangement of over-toasted RSM (RSMb) and formaldehyde-treated RSM (RSMd). This contrasts with the cereal by-products and SBP, which were also subjected to heat and pressure treatments, but clustered together regardless of the assumed ruminal passage rate. Cereal grains (barley and wheat) were allocated differently although they have similar nutrient composition (Appendix A). Matrix effects and treatments of feedstuffs seem to have a decisive influence on in vitro CP degradation as well and determine whether and to what extent protease can work. Thus, matrix effects determine the feed-specific difference between in situ and in vitro CP degradation.

Clustering was tried to identify groups of feedstuffs that should be tested with specific additional enzymes. In general, most of the feedstuffs clustered diffusely of origin and treatment, resulting in clusters that were not characterized by feedstuffs with uniform nutrient composition. The feedstuffs which were arranged outside of any cluster were not characterized by uniform nutrient composition. This makes it difficult to derive clear specific recommendations for type and quantity of the additions of carbohydrate-degrading enzymes to improve the vulnerability of the feed protein in the *S. griseus* protease test. Groupwise degradation occurred in grass silages, cereal based-byproducts together with SBP and partly in legume grains. The addition of fiber- or cell wall-degrading enzymes seems appropriate for the above-mentioned feedstuffs with exception of legume grains, to minimize differences to in situ CP degradation. The legume grains clustered, but the low degQ showed good agreement between in situ and in vitro ED (Figure 3). However, in the case of faba beans and corn, the addition of starch-degrading enzymes appears to be necessary to enable the estimation of effective CP degradation.

An objective of the present study was to assess the sensitivity of the *S. griseus* protease test displaying feed-specific treatment effects, because this is an essential requirement for feed evaluation purposes. As our results have shown, thermic, chemical and expander treatments were well discerned from the untreated materials. The differences of in situ ED (Appendix A) were in principle reflected by ED estimated in vitro (Figure 4). Thus, our results clearly confirmed that the sensitivity of the *S. griseus* protease test is reliable for evaluating specific treatment effects.

## 5. Conclusions

Results of the current study revealed that in situ CP degradation was mainly underestimated using the *S. griseus* protease test, probably due to associations of protein to carbohydrates. Feed characteristics such as nutrient composition or treatment did not fully explain the clustering of feedstuffs we observed with regard to differences between in situ and in vitro CP degradation. The clustering results do not allow a clear conclusion on the groupwise or feed-specific use of carbohydrate-degrading enzymes. The addition of amylolytic and/or fibrolytic enzymes or multi-enzyme mixtures as pre- or coincubation agents in the *S. griseus* protease test seems to be required in some cases to support starch associated and fiber-bound protein degradation. The *S. griseus* protease test displays effects of nutrient composition and treatment and could, therefore, become a reliable tool in routine feed evaluation.

## Figures and Tables

**Figure 1 animals-13-00224-f001:**
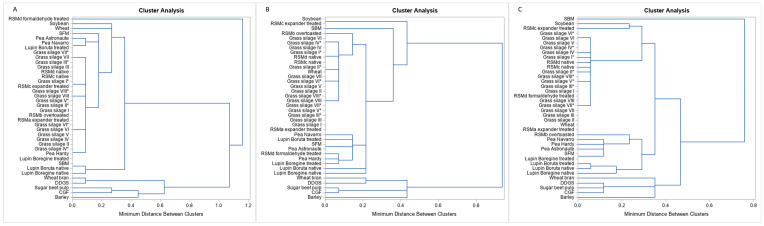
Cluster analysis according to degQ at 0.02 h^−1^ (**A**), 0.05 h^−1^ (**B**) and 0.08 h^−1^ (**C**) assumed ruminal passage rate. * GS ensiled with bacterial inoculant; CGF: corn gluten feed; DDGS: dried distillers’ grains with solubles; GS: grass silage; RSM: rapeseed meal; SBM: soybean meal; SBP: sugar beet pulp; SFM: sunflower meal.

**Figure 2 animals-13-00224-f002:**
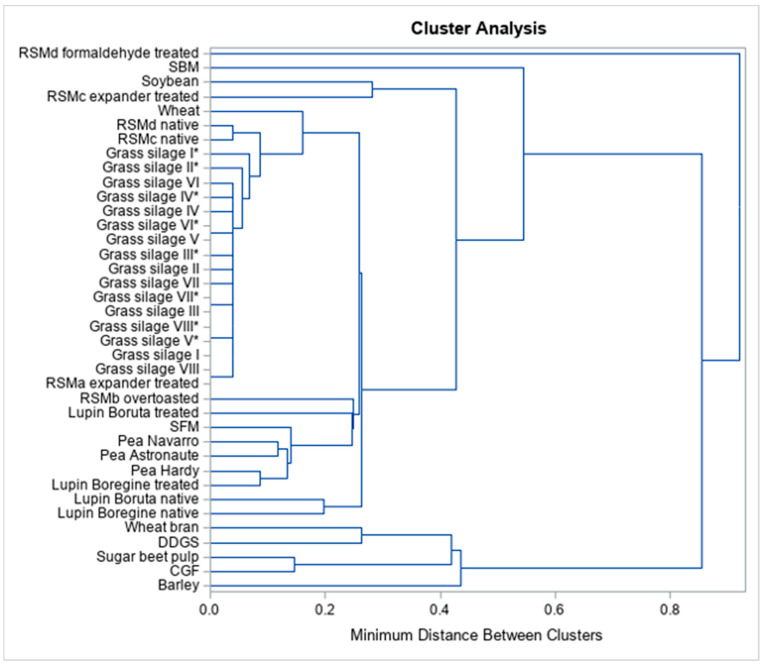
Cluster analysis including all degQ. * GS ensiled with bacterial inoculant; CGF: corn gluten feed; DDGS: dried distillers’ grains with solubles; GS: grass silage; RSM: rapeseed meal; SBM: soybean meal; SBP: sugar beet pulp; SFM: sunflower meal.

**Figure 3 animals-13-00224-f003:**
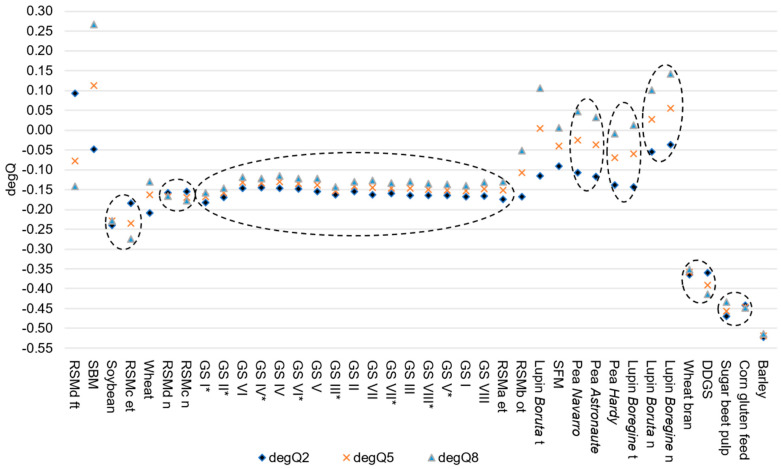
Clusters based on included degQ at 0.02 h^–1^, 0.05 h^−1^ and 0.08 h^−1^ assumed ruminal passage rate. * GS ensiled with bacterial inoculant; DDGS: dried distillers’ grains with solubles; et: expander–treated ft: formaldehyde–treated; GS: grass silage; n: native; ot: over–toasted; RSM: rapeseed meal; SBM: soybean meal; SFM: sunflower meal; t: treated. Dashed line indicates a cluster.

**Figure 4 animals-13-00224-f004:**
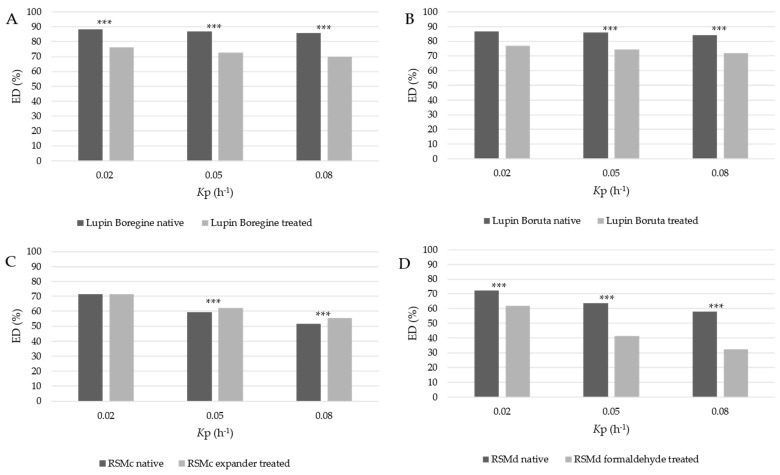
Effective crude protein degradation (ED) at 0.02 h^–1^, 0.05 h^–1^ and 0.08 h^–1^ ruminal passage rate estimated in vitro in native and treated feedstuffs: lupin *Boregine* (**A**), lupin *Boruta* (**B**), RSMc (**C**) and RSMd (**D**) (24 h incubation time). *** Asterisks indicate significant differences (*p* < 0.001); *K*p: assumed ruminal passage rate; lupin *Boregine* treated: toasted at 115–120 °C for 1 min, conditioned for 30 min in a cooling tower followed by cooling to 20 °C; lupin Boruta treated: moisture conditioning, short time toasting at 130 °C and drying to 940 g DM/kg; RSM: rapeseed meal.

**Table 1 animals-13-00224-t001:** Description of treatment procedures for lupin varieties and rapeseed meals.

Feedstuff	Treatment
Lupin *Boregine* native	Native
Lupin *Boregine* treated	Toasted at 115–120 °C for 1 min, conditioned for 30 min in cooling tower followed by cooling to 20 °C.
Lupin *Boruta* native	Native
Lupin *Boruta* treated	Moisture conditioned, short time toasted at 130 °C and drying to 940 g DM/kg.
RSMa expander-treated	Expanded (unknown conditions)
RSMb over-toasted	Toasted at 107 °C for 60 min under 450 kPa pressure [21]
RSMc native	Native
RSMc expander-treated	Expanded (unknown conditions)
RSMd native	Native
RSMd formaldehyde-treated	Formaldehyde-treated (unknown conditions)

DM: dry matter; RSM: rapeseed meal.

**Table 2 animals-13-00224-t002:** Concentration of dry matter (DM, g/kg), proximate nutrients (g/kg DM) and soluble protein (SP, % of CP) of the feedstuffs.

Feedstuff	DM	CA	CP	TP	SP	AEE	aNDFom	ADFom	Starch
Barley	894	27	125	114	28	33	162	76	532
Wheat	857	19	140	126	32	31	86	36	544
Corn	894	17	83	76	18	43	84	30	705
Wheat bran	859	59	186	163	34	55	398	135	158
DDGS	863	64	312	252	21	81	338	198	28
CGF	867	84	169	87	55	36	343	95	159
Soybean	901	56	391	373	9	221	109	65	n.a.
SBM	893	71	504	465	11	26	111	66	18
SFM	910	78	318	294	36	30	402	307	17
RSMa et	790	79	358	340	20	45	321	225	38
RSMb ot	924	85	366	352	15	23	476	269	9
RSMc n	885	86	384	357	15	46	324	210	n.a.
RSMc et	900	86	383	358	15	48	321	207	n.a.
RSMd n	900	78	374	359	24	37	324	233	n.a.
RSMd ft	911	86	370	359	9	41	339	221	n.a.
Faba bean	897	39	279	233	55	23	173	129	358
Lupin *Boregine* n	917	37	298	289	75	68	264	235	298
Lupin *Boregine* t	929	38	320	300	32	75	252	212	320
Lupin *Boruta* n	900	37	319	311	64	67	249	213	319
Lupin *Boruta* t	925	38	328	311	36	67	265	201	328
Pea *Hardy*	902	32	219	206	68	19	115	83	451
Pea *Astronaute*	901	30	228	215	70	19	118	75	432
Pea *Navarro*	898	30	248	234	73	20	142	81	392
SBP	862	85	94	57	40	19	347	174	n.a.
GS I	932	113	153	48	68	48	463	292	n.a.
GS I *	930	117	159	57	68	49	519	307	n.a.
GS II	926	102	153	50	66	40	498	293	n.a.
GS II *	917	105	156	56	65	46	503	309	n.a.
GS III	928	105	147	48	66	41	508	307	n.a.
GS III *	926	104	148	50	66	44	514	316	n.a.
GS IV	934	103	153	48	67	38	529	311	n.a.
GS IV *	937	107	155	50	66	42	520	315	n.a.
GS V	932	103	150	52	65	38	526	310	n.a.
GS V *	932	107	153	53	64	40	510	308	n.a.
GS VI	932	110	153	52	66	36	527	311	n.a.
GS VI *	929	110	153	51	64	38	518	312	n.a.
GS VII	926	108	146	55	65	34	523	307	n.a.
GS VII *	929	109	152	57	65	36	507	299	n.a.
GS VIII	927	111	151	58	65	33	526	309	n.a.
GS VIII *	932	110	149	60	65	34	514	301	n.a.

* Grass silage ensiled with bacterial inoculant; ADFom: acid detergent fiber expressed exclusive of residual ash; AEE: acid ether extract; aNDFom: neutral detergent fiber treated with amylase and expressed exclusive of residual ash; CA: crude ash; CGF: corn gluten feed; CP: crude protein; DDGS: dried distillers’ grains with solubles; et: expander–treated; ft: formaldehyde–treated; GS: grass silage; n: native; n.a.: not analyzed; ot: over–toasted; RSM: rapeseed meal; SBM: soybean meal; SBP: sugar beet pulp; SFM: sunflower meal; t: treated; TP: true protein. SP was calculated according to Licitra et al. [23] and for GS according to Higgs et al. [24]. TP was calculated as CP—non-protein nitrogen according to Licitra et al. [23].

**Table 3 animals-13-00224-t003:** Comparison of in situ (72 h incubation time) and in vitro (24 h incubation time) estimates of effective CP degradation (ED, % of CP) at 0.02 (ED_2_), 0.05 (ED_5_) and 0.08 h^−1^ (ED_8_) assumed ruminal passage rate.

Feedstuff	ED_2_		ED_5_		ED_8_	
	In Situ	In Vitro	In Situ	In Vitro	In Situ	In Vitro
Barley	91 ^aA^	43 ^bB^	87 ^aA^	42 ^bB^	83 ^aA^	40 ^bB^
Wheat	92 ^aA^	73 ^bB^	84 ^aA^	70 ^bB^	78 ^aA^	67 ^bB^
Wheat bran	91 ^aA^	58 ^bA^	87 ^aA^	56 ^bA^	83 ^aA^	54 ^bA^
DDGS	86 ^aA^	55 ^bB^	82 ^aA^	50 ^bB^	79 ^aA^	46 ^bB^
Corn gluten feed	92 ^aA^	51 ^bA^	89 ^aA^	49 ^bA^	87 ^aA^	48 ^bA^
Soybeans	92 ^aA^	70 ^bB^	84 ^aA^	65 ^bB^	79 ^aA^	61 ^bB^
SBM	84 ^aA^	80 ^bB^	68 ^aA^	75 ^bB^	56 ^aA^	71 ^bB^
SFM	89 ^aA^	81 ^bB^	81 ^aA^	77 ^bB^	74 ^aA^	75 ^aA^
RSMa et	81 ^aA^	67 ^bB^	71 ^aA^	60 ^bB^	63 ^aA^	55 ^bB^
RSMb ot	70 ^aA^	58 ^bB^	58 ^aA^	52 ^bB^	50 ^A^	47 ^B^
RSMc n	85 ^aA^	71 ^bA^	72 ^aA^	59 ^bA^	63 ^aA^	52 ^bA^
RSMc et	88 ^A^	72 ^A^	81 ^A^	62 ^A^	76 ^aA^	55 ^bA^
RSMd n	86 ^aA^	72 ^bA^	77 ^aA^	64 ^bA^	70 ^aA^	58 ^bA^
RSMd ft	57 ^A^	62 ^A^	45 ^aA^	41 ^aA^	38 ^A^	32 ^A^
Lupin *Boregine* n	92 ^aA^	88 ^bB^	82 ^aA^	87 ^bB^	75 ^aA^	86 ^bB^
Lupin *Boregine* t	89 ^aA^	76 ^bB^	77 ^aA^	73 ^bA^	69 ^aA^	70 ^aA^
Lupin *Boruta* n	92 ^aA^	87 ^bB^	83 ^aA^	86 ^aA^	77 ^aA^	84 ^bB^
Lupin *Boruta* t	87 ^aA^	77 ^bB^	74 ^aA^	74 ^aA^	65 ^aA^	72 ^bB^
Pea *Hardy*	93 ^aA^	80 ^bB^	85 ^aA^	79 ^bB^	79 ^aA^	78 ^aA^
Pea *Astronaute*	92 ^aA^	81 ^bA^	84 ^aA^	81 ^aA^	77 ^aA^	80 ^aA^
Pea *Navarro*	92 ^A^	82 ^A^	83 ^aA^	81 ^aA^	76 ^aA^	80 ^bA^
Sugar beet pulp	89 ^aA^	47 ^bA^	77 ^aA^	42 ^bA^	69 ^aA^	39 ^bA^
Grass Silage I	94 ^aA^	78 ^aA^	91 ^aA^	77 ^bA^	89 ^aA^	76 ^bA^
Grass Silage I *	94 ^aA^	77 ^bA^	91 ^aA^	75 ^bA^	88 ^aA^	74 ^bA^
Grass Silage II	93 ^aA^	79 ^bA^	90 ^aA^	77 ^bA^	88 ^aA^	76 ^bA^
Grass Silage II *	93 ^aA^	77 ^bA^	89 ^aA^	75 ^bA^	86 ^aA^	74 ^bA^
Grass Silage III	93 ^aA^	78 ^bA^	89 ^aA^	76 ^bA^	87 ^aA^	75 ^bA^
Grass Silage III *	93 ^aA^	78 ^bA^	89 ^aA^	76 ^bA^	87 ^aA^	74 ^bA^
Grass Silage IV	93 ^aA^	79 ^bA^	89 ^aA^	78 ^bA^	87 ^aA^	77 ^bA^
Grass Silage IV *	92 ^aA^	79 ^bA^	89 ^aA^	77 ^bA^	86 ^aA^	76 ^bA^
Grass Silage V	92 ^aA^	78 ^bA^	88 ^aA^	76 ^bA^	86 ^aA^	75 ^bA^
Grass Silage V *	92 ^A^	77 ^A^	88 ^aA^	75 ^bA^	86 ^aA^	74 ^bA^
Grass Silage VI	93 ^aA^	79 ^bA^	89 ^aA^	77 ^bA^	86 ^aA^	76 ^bA^
Grass Silage VI *	93 ^aA^	79 ^bA^	89 ^aA^	77 ^bA^	86 ^aA^	75 ^bA^
Grass Silage VII	92 ^aA^	77 ^bA^	87 ^aA^	75 ^bA^	84 ^aA^	73 ^bA^
Grass Silage VII *	92 ^aA^	78 ^bA^	88 ^aA^	75 ^bA^	85 ^aA^	73 ^bA^
Grass Silage VIII	92 ^aA^	77 ^bA^	87 ^aA^	75 ^bA^	84 ^A^	73 ^A^
Grass Silage VIII *	92 ^aA^	77 ^bA^	87 ^aA^	74 ^bA^	83 ^aA^	72 ^bA^
Range of SD	0.07–16.38	0.29–3.68	0.31–9.22	0.24–3.16	0.16–5.8	0.17–3.70

* Grass silage ensiled with bacterial inoculant; ^ab^ different lowercase superscripts mark significant differences with *t*-test between in situ and in vitro ED (*p* < 0.05); ^AB^ different uppercase superscripts mark significant differences with Wilcoxon rank sum test between in situ and in vitro ED (*p* < 0.05); CP: crude protein; DDGS: dried distillers’ grains with solubles; et: expander–treated; ft: formaldehyde–treated; n: native; ot: over–toasted; RSM: rapeseed meal; SBM: soybean meal; SFM: sunflower meal; t: treated. The in situ CP degradation data were corrected for the amount of microbial nitrogen present in the feed residues at each specific incubation time using the equations of Parand and Spek [6].

**Table 4 animals-13-00224-t004:** Degradation quotient (degQ) at 0.02 h^−1^, 0.05 h^−1^ and 0.08 h^−1^ assumed ruminal passage rates.

Feedstuffs	0.02 h^−1^	0.05 h^−1^	0.08 h^−1^
Barley	−0.52	−0.52	−0.51
Wheat	−0.21	−0.16	−0.13
Wheat bran	−0.37	−0.36	−0.35
DDGS	−0.36	−0.39	−0.41
CGF	−0.44	−0.45	−0.45
Soybeans	−0.24	−0.23	−0.23
SBM	−0.05	0.11	0.27
SFM	−0.09	−0.04	0.01
RSMa et	−0.17	−0.15	−0.13
RSMb ot	−0.17	−0.11	−0.05
RSMc n	−0.16	−0.17	−0.18
RSMc et	−0.18	−0.23	−0.27
RSMd n	−0.16	−0.17	−0.17
RSMd ft	0.09	−0.08	−0.14
Lupin *Boregine* n	−0.04	0.06	0.14
Lupin *Boregine* t	−0.14	−0.06	0.01
Lupin *Boruta* n	−0.05	0.03	0.10
Lupin *Boruta* t	−0.11	0.00	0.11
Pea *Hardy*	−0.14	−0.07	−0.01
Pea *Astronaute*	−0.12	−0.04	0.03
Pea *Navarro*	−0.11	−0.02	0.05
Sugar beet pulp	−0.47	−0.46	−0.43
GS I	−0.17	−0.15	−0.14
GS I *	−0.18	−0.17	−0.16
GS II	−0.15	−0.14	−0.13
GS II *	−0.17	−0.16	−0.15
GS III	−0.16	−0.15	−0.13
GS III *	−0.16	−0.15	−0.14
GS IV	−0.15	−0.13	−0.11
GS IV *	−0.14	−0.13	−0.12
GS V	−0.15	−0.14	−0.12
GS V *	−0.17	−0.15	−0.14
GS VI	−0.15	−0.13	−0.12
GS VI *	−0.15	−0.14	−0.12
GS VII	−0.16	−0.14	−0.13
GS VII *	−0.16	−0.15	−0.13
GS VIII	−0.17	−0.15	−0.13
GS VIII *	−0.17	−0.15	−0.14

* Grass silage ensiled with bacterial inoculant; CGF: corn gluten feed; DDGS: dried distillers’ grains with solubles; et: expander–treated; ft: formaldehyde–treated; n: native; ot: over–toasted; t: treated; RSM: rapeseed meal; SBM: soybean meal; SFM: sunflower meal.

## Data Availability

The data presented in this study are available on request from the corresponding author.

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
