# Peer review of "Feed Clusters According to In Situ and In Vitro Ruminal Crude Protein Degradation"

_animals, 2023, doi:10.3390/ani13020224_

Round 1

Reviewer 1 Report

The article presents a great contribution to the characterization of the use of feed for ruminant animals. Knowing the degradation of the different sources allows you to more adequately adjust the feeds to be supplied to the animals. Another important aspect is the validation of the in vitro technique with reference to in situ, which allows its use in time of the need to reduce costs and issues with ethics in the use of animals.

Reviewer 2 Report

The paper contains valuable data. Results were properly reported, and the findings have been accurately discussed and compared with other published papers. For further improvement of the manuscript, it requires some modification.

P2,L53 = Introduction

The introduction needs to be entirely re-written. It is very vague, and does not give the reader the necessary context to understand why you used the treatments you did, and why you made the measurements that you did. 

You can use new references such as:

Palangi, V., & Macit, M. (2019). In situ crude protein and dry matter ruminal degradability of heat-treated barley. Rev Méd Vét, 170, 123-128.

Eslampeivand, A., Taghizadeh, A., Safamehr, A., Palangi, V., Paya, H., Shirmohammadi, S., ... & Abachi, S. (2022). Nutritive value assessment of orange pulp ensiled with urea using gas production and nylon bag techniques. Biomass Conversion and Biorefinery, 1-9.

P4,L101 = Materials and Methods

Please move the experimental animal ethics committee protocol no.

Given that you have calculated the ED with SAS software, write the program code.

P8,L251 = Results

The in vitro ED is much lower than in situ method, why?

Under normal conditions, increasing the ruminal passage rates leads to decrease degradability, why degQ is increased in some treatments?

P15,L375 = Discussion

Based on your discussion “Although degradation values from in vivo studies”, how is the in vivo degradability calculated and what does it have relation with your results?

Based on your discussion “As a first step we examined the impact of nitrogen from increasing adherence of microbial biomass to the feed residues during in situ incubation”, do you have evidence to justify this claim? Please provide a reference.

Reviewer 3 Report

The authors have done a good job in preparing the manuscript. However, one of the main drawbacks of this manuscript is that it is less interesting to the readers from the beginning. It may be because even the simple summary looks a little complicated. Thus, make it simple and mention what you wanted to do during the study. In addition, I feel you can also improve the introduction section and shorten the discussion section focusing only on the main findings of your study. Do not try to include and justify all the findings of your study, citing the findings of others.

Please also pay attention to the followings.   

Line 79: Check for correct spaces here and elsewhere

Lines 88-89: Use one type, i.e., or, e.g., here and elsewhere

Line 95: These clusters or This cluster?

Page No. 3: Do not leave spaces between sections. Check for uniformity throughout the manuscript and remove unnecessary spaces.

Table 2: Please try to arrange tables and even figures into one page. You can make changes to font size and fit them into a page here and elsewhere.

Line 104: Any particular reason to select 40 feedstuffs?

Line 137: “in situ” should be italicized.

Lines 138-149: Please provide specifications for the bags used, for instance, manufacturer and other info. In addition, please maintain uniformity; for instance, when you start a paragraph, place a tab, and do not use one-sentence paragraphs. Check for this issue here and elsewhere, and correct them.

Line 151 and 173: Please check for spaces between a subtitle and a paragraph. Please maintain uniformity throughout.

Line 173: I would like to see the full equations used here. It would provide a better understanding to the reader rather than explaining the terms only in words.  

Line 175: “in situ” should be italicized. Please check for these issues throughout.

Line 217: Please check for correct spaces between the words, here and elsewhere.

Line 237: Please check for author guidelines for the correct way of writing “p.” Is it “p” or “P”? Correct accordingly.

Line 247: Do not start a sentence with an abbreviation. Check throughout.

Lines 532-535: What is this? I do not think this section is related to your study. Please check these issues throughout and revise accordingly.

Line 536: Your objective was “The objective of the present study was to assess the suitability of the Streptomyces griseus protease test for estimating ED of CP from 40 feedstuffs using the in-situ test as a reference method”. However, that is hardly reflected in the conclusion section. Please adhere to your main conclusion or hypothesis when you conclude your research. So, please revise this section accordingly.
